# Depressive Symptoms of Public Health Medical Residents during the COVID-19 Pandemic, a Nation-Wide Survey: The PHRASI Study

**DOI:** 10.3390/ijerph20095620

**Published:** 2023-04-24

**Authors:** Fabrizio Cedrone, Nausicaa Berselli, Lorenzo Stacchini, Valentina De Nicolò, Marta Caminiti, Angela Ancona, Giuseppa Minutolo, Clara Mazza, Claudia Cosma, Veronica Gallinoro, Alessandro Catalini, Vincenza Gianfredi

**Affiliations:** 1Hospital Management, Local Health Authority of Pescara, 65100 Pescara, Italy; 2Department of Biomedical, Metabolic and Neural Sciences, University of Modena and Reggio Emilia, Via Campi 287, 41125 Modena, Italy; 3Department of Health Sciences, University of Florence, 50134 Florence, Italy; 4Department of Public Health and Infectious Disease, Sapienza University of Rome, 00185 Rome, Italy; 5Department of Medicine and Surgery—Sector of Public Health, University of Perugia, 06100 Perugia, Italy; 6School of Hygiene and Preventive Medicine, Vita-Salute San Raffaele University, 20132 Milan, Italy; 7Department of Health Promotion, Mother and Child Care, Internal Medicine and Medical Specialties, University of Palermo, 90127 Palermo, Italy; 8Department of Public Health, Experimental and Forensic Medicine, University of Pavia, 27100 Pavia, Italy; 9Department of Biomedical Sciences and Public Health, Università Politecnica delle Marche, 60100 Ancona, Italy; 10Department of Biomedical Sciences for Health, University of Milan, Via Pascal, 36, 20133 Milan, Italy; 11CAPHRI Care and Public Health Research Institute, Maastricht University, 6211 Maastricht, The Netherlands

**Keywords:** mental health, depression, Patient Health Questionnaire, job satisfaction, schools, public health, healthcare personnel, cross-sectional design

## Abstract

Depression is a widespread condition, which increased during the COVID-19 pandemic among healthcare workers as well. The large workload of the pandemic response also affected Public Health Residents (PHRs) who played an important role in infection prevention and control activities. This work aims to assess depression in Italian PHRs, based on data collected through the PHRASI (Public Health Residents’ Anonymous Survey in Italy) study. In 2022, 379 PHRs completed the self-administered questionnaire containing Patient Health Questionnaire-9 to evaluate clinically relevant depressive symptoms (PHQ-9 ≥ 10). Multivariate logistic regression shows that the intention (aOR = 3.925, 95% CI = (2.067–7.452)) and the uncertainty (aOR = 4.949, 95% CI = (1.872–13.086)) of repeating the test to enter another postgraduate school/general practitioner course and the simultaneous attendance of two traineeships (aOR = 1.832, 95% CI = (1.010–3.324)) are positively related with depressive symptoms. Conversely, the willingness to work in the current traineeship place (aOR = 0.456, 95% CI = (0.283–0.734)) emerged as a protective factor. Similar results were obtained considering mild-to-severe (PHQ-9 ≥ 5) depressive symptoms and/or stratifying by sex. The findings, suggesting the protective role of job satisfaction toward depression, might entail future interventions to improve the learning experience and promote work-life balance.

## 1. Introduction

In 2019, thus before the pandemic, an estimated 970 million people worldwide suffered from a mental disorder, accounting for 13% of the world’s population, almost equally distributed among the two sexes (52% women and 48% among men). However, the prevalence of each mental disorder varies according to gender and age, despite the fact that anxiety and depression are the two most common disorders in both men and women. In particular, in 2019 anxiety and depression affected approximately 301 million people and 280 million people, respectively. Among them, 58 million children and adolescents were affected by anxiety, and 23 million by depression [1].

The COVID-19 pandemic highly impacted mental health. Depression and anxiety increased by 25% in the first year of the pandemic, accounting for approximately one billion people suffering from depression worldwide [2]. In Italy, PASSI (Progressi delle Aziende Sanitarie per la Salute in Italia; Progress by local health units toward a healthier Italy) Surveillance System regularly collects information from the Italian adult population on lifestyles and behavioral risk factors associated with the onset of chronic diseases. Data from PASSI showed a rise in the population suffering from depression symptoms (from 6% in 2016–2019 to 6.6% in 2020–2021) [3]. The negative consequences of the COVID-19 pandemic on mental disorders and psychological distress not only affected the general population but also vulnerable people [4] and healthcare workers [5]. The reason behind the high impact of the COVID-19 pandemic on people’s mental health is multifactorial. An important role has been played by the restrictive measures put in place, but also the financial insecurity, as well as the increase of a massive infodemic contributed to this phenomenon. All these factors triggered a global crisis for the mental health of the general population [6].

More in detail, women experienced a higher increase in mental health disorders during the COVID-19 pandemic, when compared to men, probably because women were more likely to be affected by the social and economic consequences of the pandemic [7]. For example, at the global level, it emerged that the increase in the prevalence of major depressive disorder and anxiety disorders among women was greater than in men, with 35.5 million additional cases in women versus 17.7 million additional cases in men for major depressive disorder and 51.8 million additional cases in women versus 24.4 million additional cases in men for anxiety disorders. Notably, pregnant women were shown to be affected by the highest rates of depression [8,9,10]. The predominance of women affected by depressive symptoms was also assessed in Italy by the PASSI Surveillance System, confirming the trend with 8.7% of Italian women suffering from depression compared to 4.5% of men [3]. There was also a greater variation in prevalence among younger age groups than among older ones, potentially reflecting the profound impact of school closures and social restrictions on young people’s mental health. In fact, for example, college students reported the highest rates of depression, anxiety, and insomnia [8,9,10].

Last, but not least, also healthcare workers suffered from mental health disorders during the COVID-19 pandemic. As a matter of fact, healthcare workers had to face a completely new infectious disease, for which no specific preventive strategies, as well as therapeutic treatments, were available [11]. Moreover, due to the novelty of the virus, the infection rapidly spread worldwide simultaneously affecting thousands of people [12]. In this scenario, healthcare workers had to manage an extreme work situation, rapid adaptations to intensive critical care situations, unprecedented numbers of critically ill patients, numerous patient deaths, and infection risks [13]. As a consequence, an extraordinary number of cases of depression, people with depressive symptoms, people with anxiety symptoms, and post-traumatic stress disorders were reported among healthcare personnel [14]. Despite the high impact of the pandemic on healthcare workers’ mental health, a few studies have been conducted so far, especially in Italy. Moreover, most of the previous studies mainly focus on structured healthcare workers, but none of them were conducted among Italian medical residents or, more specifically, on public health residents. Public health residents (PHRs) in Italy have worked on the contact tracing of COVID-19 cases and on the management of the pandemic at different levels such as hospital directions and local health authorities. Generally, PHRs constitute one of the youngest segments of healthcare workers, perhaps also one of the most susceptible due to their high workload and young age. As a matter of fact, medical residents are simultaneously required to deepen their theoretical knowledge and carry out practical activities with a certain degree of autonomy. Therefore, we used the data coming from the PHRASI (Public Health Residents’ Anonymous Survey in Italy) nationwide cross-sectional survey in order to (1) estimate the prevalence of depressive symptoms among the Italian PHRs; and (2) explore the association between several sociodemographic factors, with particular reference to job satisfaction indicators, and depressive symptoms in our target population. Our hypothesis is that job dissatisfaction is associated with more prominent depressive symptoms. Therefore, our null hypothesis is that there is no correlation between job satisfaction and depressive symptoms.

## 2. Materials and Methods

Public Health Residents’ Anonymous Survey in Italy (PHRASI) is a nationwide cross-sectional study conducted among Italian Public Health Medical Residents to investigate several dimensions of their mental health and its determinants. Methodological aspects have been previously detailed [15]. In brief, it is a voluntary-based, self-administered, electronic survey developed on Google Form (©2022 Google, Mountain View, CA, USA) and targeted to all PHRs (approximately 1600) enrolled in the four-year-course of any of the Italian postgraduate Public Health schools. The minimum sample size needed was calculated using the formula provided by Charan and Biswas for cross-sectional studies: a sample size of 315 was calculated [16]. All data were self-reported. The compulsory compilation option for all 88 items of the questionnaire was enabled on Google Forms in order not to have missing data. The survey link was spread through the mailing list of the medical residents’ Assembly of the Italian Society of Hygiene and Preventive Medicine. Moreover, the representatives of each postgraduate Public Health school were contacted to ask to spread the survey among their colleagues.

Data collection started on 14 June 2022 and ended on 26 July of the same year. All data collected were recorded on a computerized anonymous database, and the file was protected by a password, known only to the researchers. Since questionnaire data were anonymous, making it impossible to identify any responder, this study did not require the approval of an Ethic Committee. In fact, the answers received were analyzed only aggregately and by observing the Italian and European laws about personal data management [17,18,19].

### 2.1. Variables of Interest

The sociodemographic characteristics considered for this analysis were: age, sex, region of residence, cohabitation, number of children, region of the traineeship, course year in the postgraduate school, whether off-site, whether the commuter, willingness to work in the current work/training place after completion of the postgraduate course, simultaneous attendance of two traineeships, intention to repeat the test to enter a different postgraduate school/general practitioner course, having a contract of employment compatible with the postgraduate school, possibility to make ends meet with their own income. Age was processed as a continuous variable. The “Number of children” variable was categorized as follows: “0” for no children, “1” for “one child” and “>1” for “more than one child”. Region of residence and region of traineeship were categorized into “North”, “Center” and “South and Islands” according to the Italian National Institute of Statistics (ISTAT). In detail, “North” includes the following regions: Aosta Valley, Liguria, Lombardy, Piedmont, Emilia-Romagna, Friuli-Venezia Giulia, Trentino-Alto Adige/Südtirol, and Veneto; “Center” includes: Lazio, Marche, Tuscany, and Umbria; lastly, “South and Islands” includes: Abruzzo, Apulia, Basilicata, Calabria, Campania, Molise, Sardinia, and Sicily. The remaining variables were dichotomized as follows: cohabitation was divided into “people who live alone” or “live with others”, regardless of living with roommates or family members; course years in the postgraduate school was split into first and second biennium; willingness to work in the current work/training place after completion of the postgraduate course was divided in “yes” for those who answered “absolutely yes” or “more yes than no”, and “no” for the two opposite answers; “having a contract of employment compatible with the postgraduate school” variable was dichotomized as “yes” or “no” regardless of the specific type of contract indicated by the participant; “possibility to make ends meet with their own income” variable was dichotomized as “easily” for the “easily” answer and “hardly” for “with someone’s help” or “with difficulties” answers.

### 2.2. Depressive Symptoms

Depressive symptom severity was assessed with the Italian version of the Patient Health Questionnaire-9 (PHQ-9) [20]. The PHQ-9 score was computed according to the literature [21]. It comprises nine items rated on a four-point scale, from 0 (“not at all”) to 3 (“nearly every day”). Points from each item are added together to calculate a continuous total score ranging from 0 (no symptoms) to 27 (all symptoms present nearly every day). For the scope of this study, the analyses reported below were repeated two times considering both a cut-off score of 10 and a cut-off score of 5. Specifically, a PHQ-9 score of 10 or greater indicates clinically relevant (moderate to severe) depressive symptoms, while a score less than 10 rarely occurs in people with major depression and is suggestive of no, minimal or mild depressive symptoms. Instead, a PHQ-9 score of 5 or greater indicates the presence of depressive symptoms (mild to severe), compared with no or minimal depressive symptoms for scores under 5 [22].

### 2.3. Statistical Analysis

Dichotomic and categorical variables were presented as both absolute frequencies and percentages. Continuous variables were reported as mean and standard deviation (SD) for normal distribution or median and interquartile range (IQR) for non-normal distribution. Chi-square or Fisher’s exact test, as appropriate, was performed to evaluate the relationship between the two-level depression variable and the socio-demographic, working/training, and financial status. Student’s t-test or Wilcoxon–Mann–Whitney test, as appropriate, was performed to evaluate the difference in the distribution of continuous variables between the two groups. To assess the difference between groups with more than two groups, ANOVA or Kruskal–Wallis test, as appropriate, was performed.

Multicollinearity between independent variables was assessed with Pearson’s correlation coefficient [23]. A Pearson’s coefficient of 0.7 or greater among two or more predictors indicates the presence of multicollinearity. Multiple logistic regressions adjusted for sex and age were performed. Adjusted Odds Ratio (aOR) and their relative 95% confidence interval (95% CI) were reported. Furthermore, multiple negative binomial regressions adjusted for sex and age were performed using the PHQ-9 score as a continuous variable. Adjusted Risk Ratios (aRR) and their relative 95% CI were reported. *p*-value ≤ 0.05 was considered statistically significant. All the analyses were performed using R 4.2.2 [24].

## 3. Results

The characteristics of the sample are summarized in Table 1. Regarding the study population, 58% of included patients are females (219 subjects) and 42% are males (160 subjects). The mean age of the sample is 30 years, with a range between 25 and 61 years (median age of 29 and IQR of 34 years). Considering the region of residence, 41% of the participants (157 people) lived in northern Italy, 25% (96 people) in central Italy, and 33% (126 people) in southern Italy and in the islands. Overall, 61% (231 people) of the sample have depressive symptoms (PHQ-9 ≥ 5, Appendix A), whereas 26% of the sample (97 people) have clinically relevant depressive symptoms (PHQ-9 ≥ 10, Table 1). Residents with clinically relevant depressive symptoms (PHQ-9 ≥ 10) have a less favorable job-satisfaction profile compared to those without: they are less willing to work in their current work/training place after completion of the postgraduate course, more prone to repeat the test to enter a different postgraduate school/general practitioner course and more frequently attend two traineeships simultaneously (Table 1). Similar results were found when mild-to-severe (PHQ-9 ≥ 5) depressive symptoms were considered (Appendix A).

Multicollinearity analysis revealed no collinearity among the independent variables (Appendix A). In fact, all the variables, even if significantly associated, had a Pearson’s correlation coefficient (r) < 0.70.

The multiple logistic regressions (in Table 2) show that the intention and the uncertainty of repeating the test to enter a different postgraduate school or the general practitioner course were the factors most associated with clinically relevant (PHQ-9 ≥ 10) depressive symptoms (aOR = 3.925, 95% CI = (2.067–7.452), *p*-value = 0.001 and aOR = 4.949, 95% CI = (1.872–13.086), *p*-value < 0.001, respectively) compared to those who do not intend to repeat the test. The willingness to work in the current work/training place after completion of the postgraduate course was inversely associated with clinically relevant depressive symptoms (aOR = 0.456, 95% CI = (0.283–0.734), *p*-value = 0.001), compared with the lack of willingness of repeating the test. The simultaneous attendance of two traineeships was associated with clinically relevant depressive symptoms (aOR = 1.832, 95% CI = (1.010–3.324), *p*-value = 0.049), compared to attending only one traineeship at a time. Similar results were obtained when mild-to-severe (PHQ-9 ≥ 5) depressive symptoms were considered (Appendix A).

Stratifying the results by sex (Table 3), the intention to repeat the test to enter a different postgraduate school/general practitioner course was strongly related to clinically relevant (PHQ-9 ≥ 10) depressive symptoms, both in males and females. Similar results were found considering mild-to-severe (PHQ-9 ≥ 5) depressive symptoms (Appendix A). The unwillingness to work in the current work/training place after completion of the postgraduate course was associated with clinically relevant (PHQ-9 ≥ 10) depressive symptoms in females. On the other hand, the simultaneous attendance of two traineeships was associated with clinically relevant depressive symptoms in males. When mild-to-severe (PHQ-9 ≥ 5) depressive symptoms were considered, being a commuter worker was associated with depressive symptoms in males, while hardly making ends meet was associated with depressive symptoms in females (Appendix A).

Lastly, in order to increase the robustness of our results, we considered PHQ-9 as a continuous variable (Appendix A). Statistically significant results did not materially change. More in detail, the willingness to work in the current work/training place was inversely associated with more depressive symptoms on a continuous scale (aRR = 0.754; 95% CI = (0.646–0.882), *p*-value < 0.001). On the contrary, the intention (aRR = 1.677; 95% CI = (1.212–2.321), *p*-value = 0.002) and the uncertainty (aRR = 1.686; 95% CI = (1.364–2.085), *p*-value < 0.001) of repeating the postgraduate school/general practitioner test, and the simultaneous attendance of two traineeships (aRR = 1.288; 95% CI = (1.051–1.578), *p*-value = 0.015) were again associated with more depressive symptoms on a continuous scale.

## 4. Discussion

This study aims to assess the prevalence of self-reported depressive symptoms during the COVID-19 pandemic among PHRs in Italy as well as their association with a broad range of socio-demographic characteristics. Our data contribute to building a body of evidence on the psychiatric burden of diseases in the medical residents’ population and reveal that depression is a common mental disorder also in PHRs. In fact, 1 in 4 (26%, n = 97 people), showed clinically relevant depressive symptoms. It should be considered that PHRs are at the same time students and workers, investing most of their energy and time in their residency. This is presumably the reason why some of the work-related issues explored in the current study have been significantly associated with depressive symptoms.

Our results are in line with previous research assessing the association between socio-demographic factors and depressive symptoms in medical personnel at the early career stage, as residents and medical students. According to a national survey [25] conducted in the United States of America (USA), residents and medical students are more likely to exhibit symptoms of depression than the general population; a systematic review in 2014 [26] confirmed that the prevalence of depression in residents is higher than in the general population. Factors that can be associated with residents’ depression are the decreased ability to handle work-related stress, discontinuation of medical training, disruption in personal lives, lack of access to wellness resources, lack of time off for leisure, and fewer hours of sleep [26,27]. In particular, a recent meta-analysis showed that the pooled prevalence of depression among residents ranges from 20.8% to 43.2% with a significant increase rate over time [28]. However, the most significant increase seems to have been recorded in the COVID-19 pandemic scenario, as shown, for example, in Mendonça and colleagues’ study among Brazilian residents, in which the prevalence of depression registered an approximately three-fold increase in 2021 [29].

The reasons behind this high prevalence of depression in this sub-population could be the precarious and/or unstable work situations lived by residents, as well as the lack of job satisfaction. In our analysis, we found that PHRs willing to repeat the admission test to enter other postgraduate schools or the general practitioner course and residents attending simultaneous traineeships have a higher risk of clinically relevant depressive symptoms, while the willingness to work in the current work/training place after completion of the postgraduate course resulted as a negative predictor for depressive symptoms. These aspects, closely related to job satisfaction, are in line with the study of Allan et al., in which poor job satisfaction has been associated with a high risk of depression and stress [30]. On the contrary, people perceiving their work as meaningful and satisfying reported less anxiety and stress. Moreover, a recent systematic review that aimed at linking work to the development of common mental health problems identified occupational uncertainty as one of the main work-related factors that can contribute to the development of depression and/or anxiety [31].

Applying the gender lens, according to the literature, women are subject to a higher risk of depression and related symptoms compared to men worldwide [32]. Furthermore, we know that female physicians more frequently experience additional stress struggling with gender-based discrimination and harassment, family-career balance, and salary inequity [33,34]. A recent meta-analysis shows that female healthcare workers are at a higher risk of suffering from depressive symptoms compared to males [35]. These are the reasons why the multivariate logistic regression analysis was corrected for sex. Furthermore, relations between socio-demographic characteristics and depression symptoms stratified by sex were explored. In our study, factors associated with clinically relevant depressive symptoms among female PHRs were the possibility to make ends meet with their own salary and the willingness to work in the current work/training place after completion of the postgraduate course. On the other hand, being a commuter worker and performing simultaneous internships were conditions associated with depressive symptoms in male PHRs. The intention to repeat the test to enter a different postgraduate school/general practitioner course was strongly related to clinically relevant depressive symptoms, both in males and females.

More in-depth, being a commuter worker and performing simultaneous internships are conditions associated with depressive symptoms in male residents. Being a commuter worker can have an impact on people’s mental health: longer commuting time is significantly associated with shorter working hours, less physical exercise, and shorter sleeping hours [36]. A Norwegian study on business travelers’ health found the association between emotional exhaustion and musculoskeletal pain to be significantly stronger for commuter workers compared to national and international travel groups [37].

In our analysis, the possibility of making ends meet with one’s own income was a factor associated with clinically relevant depressive symptoms among female residents. Several studies have found that depressive symptomatology is more prevalent at lower levels of socioeconomic status [38]. In the USA, the prevalence of depression is higher for those in the lowest income groups, and having a higher income is associated with reductions in psychological distress [39,40]. In Greece, a significant association was recorded between major depression and economic hardship [41]; moreover, individuals with financial distress displayed increased odds of suffering from major depression. According to a review by Ridley et al., people with the lowest incomes are typically 1.5 to 3 times more likely than the richer to experience depression or anxiety [42]. However, the opposite is also true: depression, often determining reduced productivity and different social functioning, can impact working life and lead to greater job instability compared to non-depressed individuals [43].

Financial stability can favor family formation, a significant milestone for many individuals [44]: for example, earnings were found to have an important influence on marriage timing [45]. Furthermore, the social distancing needed for the COVID-19 pandemic showed that living alone can strongly affect mental health [46]. For these reasons we expected cohabitation to be a protective factor for depressive symptoms, but we did not observe a significant association in this regard. This could be related to the relatively small dimension of our sample and the reduced statistical power that stems from this issue. Dorrit Posel is the first to investigate the mental health implications of solo-living among working-age adults in a developing country context, showing that living alone was associated with an increased risk of depression, whereas, lower depression scores were observed among adults more socially integrated [47]. People living alone would be more vulnerable to depression, suffering more from social isolation than individuals who live with others. However, it was not only living alone that predisposed them to depression. Above all, social network characteristics, such as less emotional support and a lower proportion of family members, were demonstrated to be associated with prevalent and incident depressive symptoms [48]. In fact, a more supportive social network (strong relationships and high social integration) among workers substantially reduced if not eliminated their vulnerability to depression [49].

Other sociodemographic data, including age, marital status, and raising children, had no statistically significant impact on the prevalence of depression in contrast to other studies, in which, for example, being married was independently associated with lower odds of having major depression and recent suicidal ideation [25].

### 4.1. Limitations and Strengths

Before generalizing our results, some limitations should be considered. Firstly, data collected might be prone to social desirability bias, or recall bias, particularly because this study used self-reported measures [50]. Actually, despite the fact that anonymity was guaranteed, a fear of being recognized may have impacted data reporting. However, performing additional sensitivity analyses (considering not only PHQ-9 ≥ 10 but also PHQ-9 ≥ 5 and PHQ-9 as a continuous score), the results did not materially change. Secondly, residents more isolated and more unsatisfied with their postgraduate course may have been less motivated to participate in the survey. This would result in an underestimation of both the PHRs’ depression symptoms’ prevalence and the association between job dissatisfaction characteristics and depressive symptoms.

Moreover, although the PHQ-9 questionnaire is a validated instrument to detect depressive symptoms, it is considered a screening test and does not allow a clinical diagnosis of depressive disorders. Nevertheless, it is a valid instrument, widely used, with high sensitivity and specificity. Therefore, even if clinically relevant depressive symptoms do not equate with major depressive disorder, misclassification is expected to be low.

Lastly, we used a cross-sectional design and hence, assessing cause and effect between the variables is not possible. Temporality nor causality cannot be inferred, but only the prevalence can be calculated.

Despite these limitations, the study has some strengths. To the best of our knowledge, this was the first study to investigate PHRs’ depressive symptoms in Italy and Europe and it could be considered singular and original. A major strength of this study is the sample representativeness and large size [51]. Our survey yielded high nationwide participation, with an almost equal distribution among the north, center, and south of Italy. Moreover, a much larger sample size, compared to the minimum calculated with the formula provided by Charan and Biswas, was reached [16]. Furthermore, despite not taking any control over the possibility of filling the questionnaire twice by the same person, or even if the participants were residents in another medical specialty, we tailored questions specifically for the PHRs category and disseminated the link through mailing lists and group chats in which only PHRs were present. In addition, the accompanying message to the link and the cover letter always specified that the survey was addressed to PHRs and not to residents belonging to other disciplines’ postgraduate schools. Moreover, the cover letter also provided in-depth details about how to fill in the questionnaire, and an email address, dedicated to technical and information support, was available for giving additional instructions and to assist participants, if requested.

The study has been conducted at a low cost and in a short period of time, and with no missing data. Moreover, we used validated instruments, such as the PHQ-9 questionnaire to assess depressive symptoms. This tool can be considered a valid and validated instrument, with high sensitivity and specificity, therefore we are confident that the misclassification and mismatch rates are low [52].

Additionally, for what concerns the statistical analysis, the study of the multicollinearity between the independent variables with the calculation of Pearson’s correlation coefficient makes our results more solid. Lastly, by adopting a gender-based approach, this research contributes to creating evidence on gender medicine and might have a role in understanding how depressive disorder differs between men and women in terms of socio-demographic characteristics.

### 4.2. Implications for Policy, Practice, and Research

Study findings are consistent with prior evidence suggesting that feeling unsatisfied with one’s work could result in psychological distress and mental health problems. Previously, Ogunnubi et al. found residency training to be stressful for doctors and often compounded by low levels of job satisfaction and perceived stress [53]. In support of this, our findings revealed a linear relationship between PHRs’ job satisfaction and depression, highlighting the importance of creating work-life satisfaction and a positive work environment as concrete strategies to improve residents’ gratification and mental well-being.

The COVID-19 pandemic has increased work-related stress, especially among healthcare workers [54]. In this respect, mental health screening and work environment strategies are needed to be implemented now more than ever. To date, interventions have primarily focused on individual strategies (e.g., stress reduction, resilience strengthening, mindfulness, and cognitive processing), instead of using a public health perspective [55]. To make mental health services useful, it should be ensured that residents are knowledgeable of their presence and have access to these resources. Many residents are aware that their heavy workloads and busy schedules prevent them from investing time in caring for their mental well-being. This factor is widely reported to lead to stress and burnout in training/work. Hence, it should be guaranteed that residents have dedicated time to make free use of mental services and all institutions should hire a mental health provider, free of charge for trainees and available at off-hours [56].

Our results showed that PHRs need a certain level of support to cope with mental health issues. Residents are at elevated risk for developing depression compared to the general population; however, they are also less likely to utilize mental health services [57]. Considering this, organizational interventions on curricular, educational, and systemic factors of Public Health residency programs are most of all needed. To enhance the training environment and increase PHRs’ satisfaction with their residency, factors that can be involved in residents’ depressive symptoms’ development should be improved, such as faculty supervision and the quality of the learning experience [58].

In order to raise awareness on this problem, the Working Group on “Public Mental Health” of the medical residents’ Assembly of the Italian Society of Hygiene and Preventive Medicine could share the findings of the study with the Italian Public Health Schools Directors. Our data could help Public Health Schools’ Directors to properly estimate the impact of the residency program’s logistics and workload on the mental health of PHRs. Thanks to the evidence collected in this national study, each Director could take into consideration actions to improve or actively promote PHRs’ mental well-being at their workplace. If needed, they could consider reshaping the residency training program.

Furthermore, from an economic point of view, creating a supportive work culture that enhances individual mental health is fundamental to reducing the enormous consequences of mental health disorders, such as increased healthcare expenditures with short and long-term disability costs, absenteeism, reduction of turnover and a great productivity loss [59,60].

Making mental health services more accessible and enhancing well-being in the workplace would provide real improvements in residents’ mental health outcomes. Since the quality of care is linked to the mental state of health workers, this will consequently increase healthcare performance [61]. Nevertheless, since mental health services are under-resourced compared to other healthcare services, an important financial investment in this area is needed.

The knowledge of the vulnerability of PHRs to psychological suffering and depressive symptoms should drive Public Health practitioners and policymakers to promptly strengthen primary prevention strategies. All information gathered might be used by policymakers in framing recommendations to implement several future interventions and concrete strategies to promote mental well-being, to psychologically and psycho-socially support PHRs, and to encourage the growth of psychotherapy support centers in universities, which are currently lacking [62,63]. To make these services more accessible they should promote tools shown to be highly effective, cost-efficient, and accessible, such as tele-mental health, online mental health assessments, and self-directed mental health interventions [64,65].

Finally, professional assistance might be offered to all residents, with multidisciplinary teams composed of physicians, psychiatrists, psychologists, and social workers to provide appropriate management and varying treatment options. Physicians should be incorporated into specific outreach strategies and should play a decisive key role in screening healthcare workers for possible depression. Screening programs can detect patients at the earliest stages of depression, shortening the classic 4 year gap between depression onset and the beginning of treatment and generally increasing the likelihood of remission and treatment response [66]. Currently, the screening for depression in the primary care setting is less than 5%, hence physicians should be more involved in the early recognition of depressive symptoms and early prevention of depressive disorder [67].

This nationwide cross-sectional study fills the information gap that exists regarding residents’ mental well-being and provides useful data to support the improvement of mental health surveillance in the PHR population. A longitudinal investigation is needed to determine the temporal relationships between PHR variables, to gain a better insight into changes in depressive symptoms over the course of individual careers, and above all, to better identify effective stress management programs during residency [68]. Further studies that closely examine the complex array of PHRs’ curricular factors that could potentially influence distress are needed. Moreover, since our survey was restricted to Italy, further studies should investigate whether the results cited above are replicable in other countries and investigate if depressive symptoms are a common PHRs problem or if there is something in the Italian training courses that predispose to its development more than in other countries.

## 5. Conclusions

Being the first of its kind in Italy and Europe and ensuring nationwide generalizability, this study provides relevant information on depressive symptoms in Italian PHRs by estimating their prevalence in this category of healthcare workers. Furthermore, it identifies sociodemographic factors associated with the presence of these symptoms, such as the unwillingness to work in the current work/training place after completion of the postgraduate course, the willingness to repeat the test to enter a different postgraduate school/general practitioner course, the attendance of simultaneous traineeships, being a commuter worker and the difficulty in making ends meet.

The achievement of this awareness might entail several future interventions in Public Health residencies in order to minimize the occurrence of PHRs’ dissatisfaction, improve the learning experience and promote work-life balance. The policies and the subsequent intervention strategies for public mental health should comprehend the investment in PHRs’ dedicated mental health services able to provide closer and better access to mental health professionals.

Further research is needed to explore other domains involved in the development of depressive symptoms in this specific healthcare worker population in order to identify the resources that best support residents of diverse backgrounds. This would allow for better and more solid shaping of effective preventive strategies, mental well-being assistance, and counseling programs.

## Figures and Tables

**Table 1 ijerph-20-05620-t001:** Relations between socio-demographic characteristics and clinically relevant depressive symptoms (PHQ-9 ≥ 10).

Characteristic	n, % (N = 379)	No Clinically Relevant Depressive Symptoms (PHQ-9 < 10) N = 282 ^1^	Clinically Relevant Depressive Symptoms (PHQ-9 ≥ 10) N = 97 ^1^	*p*-Value ^2^
Age [median, (IQR)]	30.0 (29.0, 34.0)	30 (28.00, 34)	31 (29.00, 33)	0.13
Sex				
Female	219 (58%)	164 (58.16%)	55 (56.70%)	0.8
Male	160 (42%)	118 (41.84%)	42 (43.30%)	
Region of residence				
Center	96 (25%)	75 (26.60%)	21 (21.65%)	0.3
North	157 (41%)	119 (42.20%)	38 (39.18%)	
South and islands	126 (33%)	88 (31.21%)	38 (39.18%)	
Cohabitation				
Alone	98 (26%)	70 (24.82%)	28 (28.87%)	0.4
With Others	281 (74%)	212 (75.18%)	69 (71.13%)	
Number of children				
0	327 (86%)	241 (85.46%)	86 (88.66%)	0.3
1	32 (8%)	23 (8.16%)	9 (9.28%)	
>1	20 (5%)	18 (6.38%)	2 (2.06%)	
Region of traineeship				
Center	113 (30%)	84 (29.79%)	29 (29.90%)	0.6
North	178 (47%)	136 (48.23%)	42 (43.30%)	
South and islands	88 (23%)	62 (21.99%)	26 (26.80%)	
Course year in the postgraduate school				
1st biennium	292 (77%)	218 (77.30%)	74 (76.29%)	0.8
2nd biennium	87 (23%)	64 (22.70%)	23 (23.71%)	
Off-site				
No	211 (56%)	159 (56.38%)	52 (53.61%)	0.6
Yes	168 (44%)	123 (43.62%)	45 (46.39%)	
Commuter				
No	258 (68%)	197 (69.86%)	61 (62.89%)	0.2
Yes	121 (32%)	85 (30.14%)	36 (37.11%)	
Willingness to work in the current work/training place after completion of the postgraduate course				
No	125 (33%)	80 (28.37%)	45 (46.39%)	0.001
Yes	254 (67%)	202 (71.63%)	52 (53.61%)	
Simultaneous attendance of two traineeships				
No	321 (85%)	245 (86.88%)	76 (78.35%)	0.044
Yes	58 (15%)	37 (13.12%)	21 (21.65%)	
Intention to repeat the test to enter a different postgraduate school/general practitioner course				
No	315 (83%)	251 (89.01%)	64 (65.98%)	<0.001
Maybe	46 (12%)	23 (8.16%)	23 (23.71%)	
Yes	18 (4.7%)	8 (2.84%)	10 (10.31%)	
Having a contract of employment compatible with the postgraduate school				
No	242 (64%)	182 (64.54%)	60 (61.86%)	0.6
Yes	137 (36%)	100 (35.46%)	37 (38.14%)	
Possibility to make ends meet with their own income				
Easily	162 (43%)	123 (43.62%)	39 (40.21%)	0.6
Hardly	217 (57%)	159 (56.38%)	58 (59.79%)	

^1^ Median (IQR); n (%); ^2^ Wilcoxon rank sum test; Pearson’s Chi-squared test.

**Table 2 ijerph-20-05620-t002:** Multivariate logistic regression adjusted for age and sex (PHQ-9 ≥ 10).

Variable	aOR Related to Depressive Symptoms (PHQ-9 ≥ 10)	95% CI	*p*-Value
Cohabitation (ref. Alone)			
With Others	0.816	0.486–1.371	0.443
Region of traineeship (ref. Center)			
North	0.912	0.527–1.577	0.741
South and islands	1.213	0.650–2.263	0.544
Off-site (ref. No)			
Yes	1.171	0.731–1.877	0.512
Commuter (ref. No)			
Yes	1.329	0.813–2.172	0.256
Willingness to work in the current work/training place after completion of the postgraduate course (ref. No)			
Yes	0.456	0.283–0.734	0.001
Simultaneous attendance of two traineeships (ref. No)			
Yes	1.832	1.010–3.324	0.046
Intention to repeat the test to enter a different postgraduate school/general practitioner course (ref. No)			
Maybe	3.925	2.067–7.452	<0.001
Yes	4.949	1.872–13.086	0.001
Having a contract of employment compatible with the postgraduate school (ref. No)			
Yes	1.142	0.705–1.848	0.589
Possibility to make ends meet with their own income (ref. Easily)			
Hardly	1.123	0.696–1.811	0.634

**Table 3 ijerph-20-05620-t003:** Relations between socio-demographic characteristics and depression symptoms (PHQ-9 ≥ 10), stratified by sex.

Characteristic	Female			Male		
	No Clinically Relevant Depressive Symptoms (PHQ-9 < 10) N = 164 ^1^	Clinically Relevant Depressive Symptoms (PHQ-9 ≥ 10) N = 55 ^1^	*p*-Value ^2^	No Clinically Relevant Depressive Symptoms (PHQ-9 < 10) N = 118 ^1^	Clinically Relevant Depressive Symptoms (PHQ-9 ≥ 10) N = 42 ^1^	*p*-Value ^2^
Age [median, (IQR)]	30 (28.00, 34)	31 (29.00, 34)	0.10	30 (28.00, 34)	31 (29.00, 33)	0.7
Region of residence						
Center	44 (26.83%)	9 (16.36%)	0.3	31 (26.27%)	12 (28.57%)	0.4
North	72 (43.90%)	26 (47.27%)		47 (39.83%)	12 (28.57%)	
South and islands	48 (29.27%)	20 (36.36%)		40 (33.90%)	18 (42.86%)	
Cohabitation						
Alone	34 (20.73%)	15 (27.27%)	0.3	36 (30.51%)	13 (30.95%)	>0.9
With Others	130 (79.27%)	40 (72.73%)		82 (69.49%)	29 (69.05%)	
Number of children						
0	135 (82.32%)	46 (83.64%)	0.5	106 (89.83%)	40 (95.24%)	0.6
1	16 (9.76%)	7 (12.73%)		7 (5.93%)	2 (4.76%)	
>1	13 (7.93%)	2 (3.64%)		5 (4.24%)	0 (0.00%)	
Region of traineeship						
Center	48 (29.27%)	14 (25.45%)	0.9	36 (30.51%)	15 (35.71%)	0.3
North	80 (48.78%)	28 (50.91%)		56 (47.46%)	14 (33.33%)	
South and islands	36 (21.95%)	13 (23.64%)		26 (22.03%)	13 (30.95%)	
Course year in the postgraduate school						
1st biennium	123 (75.00%)	42 (76.36%)	0.8	95 (80.51%)	32 (76.19%)	0.6
2nd biennium	41 (25.00%)	13 (23.64%)		23 (19.49%)	10 (23.81%)	
Off-site						
No	86 (52.44%)	31 (56.36%)	0.6	73 (61.86%)	21 (50.00%)	0.2
Yes	78 (47.56%)	24 (43.64%)		45 (38.14%)	21 (50.00%)	
Commuter						
No	114 (69.51%)	33 (60.00%)	0.2	83 (70.34%)	28 (66.67%)	0.7
Yes	50 (30.49%)	22 (40.00%)		35 (29.66%)	14 (33.33%)	
Willingness to work in the current work/training place after completion of the postgraduate course						
No	47 (28.66%)	29 (52.73%)	0.001	33 (27.97%)	16 (38.10%)	0.2
Yes	117 (71.34%)	26 (47.27%)		85 (72.03%)	26 (61.90%)	
Simultaneous attendance of two traineeships						
No	142 (86.59%)	46 (83.64%)	0.6	103 (87.29%)	30 (71.43%)	0.018
Yes	22 (13.41%)	9 (16.36%)		15 (12.71%)	12 (28.57%)	
Intention to repeat the test to enter a different postgraduate school/general practitioner course						
No	146 (89.02%)	39 (70.91%)	0.005	105 (88.98%)	25 (59.52%)	<0.001
Maybe	13 (7.93%)	12 (21.82%)		10 (8.47%)	11 (26.19%)	
Yes	5 (3.05%)	4 (7.27%)		3 (2.54%)	6 (14.29%)	
Having a contract of employment compatible with the postgraduate school						
No	110 (67.07%)	38 (69.09%)	0.8	72 (61.02%)	22 (52.38%)	0.3
Yes	54 (32.93%)	17 (30.91%)		46 (38.98%)	20 (47.62%)	
Possibility to make ends meet with their own income						
Easily	67 (40.85%)	17 (30.91%)	0.2	56 (47.46%)	22 (52.38%)	0.6
Hardly	97 (59.15%)	38 (69.09%)		62 (52.54%)	20 (47.62%)	

^1^ Median (IQR); n (%); ^2^ Wilcoxon rank sum test; Pearson’s Chi-squared test; Fisher’s Exact Test for Count Data with simulated *p*-value (based on 10,000 replicates).

## Data Availability

Authors can be contacted for information about the data presented.

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
