# Peer review of "Depressive Symptoms of Public Health Medical Residents during the COVID-19 Pandemic, a Nation-Wide Survey: The PHRASI Study"

_ijerph, 2023, doi:10.3390/ijerph20095620_

Round 1

Reviewer 1 Report

Dear colleagues!

Your research is relevant and interesting.

I would like to ask a few questions about methodology.

1. You didn't specify a null hypothesis

2. Did you pre-calculate the sample? If so, what method and formula was used.

3. You write that you used Google forms for the survey - how did you check the sensitivity of the method and where are the guarantees of an individual approach to the survey?

As for the discussion, I have a question about the proportionality of comparing the situation in different countries. Have you made a comparison, taking into account the demographic and socio-cultural characteristics of the countries.

In conclusion, I would like to see practical recommendations that can be obtained from your research.

Author Response

Response to Reviewer 1 Comments

  1. You didn't specify a null hypothesis

We thank the Reviewer for this observation. Our hypothesis was that more prominent depressive symptoms were associated with job dissatisfaction (measured with the variables “Willingness to work in the current work/training place after completion of the postgraduate course” and “Intention to repeat the test to enter a different postgraduate school/general practitioner course”). Therefore, the null hypothesis was that there was no association between depressive symptoms and job satisfaction. We added this information in the manuscript.

  1. Did you pre-calculate the sample? If so, what method and formula was used.

We thank the Reviewer for having raised this point. We performed a sample size calculation with the formula provided by Charan and Biswas. All data regarding sample size calculation were previously published (reference no. 15, doi https://doi.org/10.3390/ijerph20032003). However, based on this suggestion, a brief description has been added to the method section of the reviewed manuscript.

  1. You write that you used Google forms for the survey - how did you check the sensitivity of the method and where are the guarantees of an individual approach to the survey?

We thank the Reviewer for having highlighted this point. Although we did not have any data regarding the sensitivity, Google Forms is largely used in research, it is free of charge and user-friendly. Moreover, even though we do not have a guarantee about the individual approach to the survey, we are confident that none of the respondents answered the questionnaire more than once. In fact, also considering the time needed to complete the questionnaire (15 minutes), we believe that none of the public health residents were interested or had time to make a double compilation. Actually, we surveyed medical doctors directly involved in COVID-19 pandemic management.

  1. As for the discussion, I have a question about the proportionality of comparing the situation in different countries. Have you made a comparison, taking into account the demographic and socio-cultural characteristics of the countries.

We thank the Reviewer for this comment. We all appreciate this and agree with the Reviewer on the importance of comparing our data with those from other countries, considering the demographic and socio-cultural aspects. Most of the data cited in the discussion come from previous systematic reviews (with meta-analyses), representing pooling data from several studies. This means that our data are compared to data from a wide geographical distribution. However, in the discussions there are also comparisons with data from single countries (USA and Greece) with demographic and socio-cultural characteristics similar to the Italian ones.

  1. In conclusion, I would like to see practical recommendations that can be obtained from your research.

We thank the Reviewer for raising this relevant aspect. The purpose of our research is to obtain data on depressive symptoms among Italian medical residents in Public Health. Despite the limitations of this study design, our results can have important practical implications, as already described in the paragraph “4.2. Implications for Policy, Practice and Research”, considering the lack of specific assistance for medical residents with depressive symptoms. In order to enlighten more practical implications of our study, we added a paragraph about the role that our data could have in reshaping Public Health Schools’ residency programs to promote PHRs’ mental health and well-being.

Reviewer 2 Report

I would like to congratulate the authors for this fruitful manuscript that discussed a hot topic about "Depressive symptoms of public health medical residents during the COVID-19"

This is an interesting study performed in Italy among a special at-risk group of "Public Health Residents (PHRs)". The study investigated assess depression in Italian PHRs, based on data collected through the PHRASI (Public Health Residents’ Anonymous Survey in Italy) study. Overall, the manuscript is well-written and the statistical analyses are precisely performed; however, I have a few comments.

Major comments:

1. Remove "Surveys and Questionnaires" from keyword lists.

2. In page 2 lines 50-21, i prefer if you add some facts from published studies during COVID-19 era; Facts in this Systematic review paper could help and need to be cited properly https://doi.org/10.1002/pchj.516

https://doi.org/10.1038/s41380-022-01638-z

3. You stated "The compulsory compilation option for all the 88 items of the questionnaire was enabled on Google Form in order not to have missing data". Why this is compulsory? what is the autonomy role in this statement?

4. The sample size calculation was poor. I would, however, like to suggest the author conduct the study on a larger sample size rather than a small sample size. Also, how the sampling was done has not been mentioned. I would recommend the author make the method more robust and detailed.

7. The authors need to present more robustly on the data collection procedure. Please clarify who was assigned to survey the study participants. Were they trained personnel or people with no medical background?

5. The conclusion lacks the implications of the findings. Please make the necessary implications of your findings and discuss the implications of this study to reshape the residency training program.

6. There are numerous grammar and spelling errors throughout the manuscript that raise questions regarding the quality of the manuscript's preparation. Therefore, I request the authors screen for all grammatical and spelling errors and revise the manuscript text accordingly.

After addressing these issues, the manuscript is good for publication and is the first report among a special at-risk group of "Public Health Residents (PHRs)"

Author Response

Response to Reviewer 2 Comments

  1. Remove "Surveys and Questionnaires" from keyword lists.

We thank the Reviewer for this suggestion. We have removed the “Surveys and Questionnaires" keyword. Instead, we added “Job Satisfaction” as a further and appropriate keyword for our research.

  1. In page 2 lines 50-21, i prefer if you add some facts from published studies during COVID-19 era; Facts in this Systematic review paper could help and need to be cited properly https://doi.org/10.1002/pchj.516https://doi.org/10.1038/s41380-022-01638-z

Thank you very much for the valuable suggestion. The manuscript has been changed as suggested

  1. You stated "The compulsory compilation option for all the 88 items of the questionnaire was enabled on Google Form in order not to have missing data". Why this is compulsory? what is the autonomy role in this statement?

We thank the Reviewer for this comment. The reason behind compulsory answer setting is linked to avoiding missing data. It means that, before proceeding in filling in the next question, all the previous questions should be answered. It does not imply that the respondent should reply against his/her will. Actually, participants were free to withdraw from the survey at any time. The compulsory setting is most useful in case some previous questions were not filled in because of forgetfulness.

  1. The sample size calculation was poor. I would, however, like to suggest the author conduct the study on a larger sample size rather than a small sample size. Also, how the sampling was done has not been mentioned. I would recommend the author make the method more robust and detailed.

We thank the Reviewer for having raised this point. We performed a sample size calculation with the formula provided by Charan and Biswas. All data regarding sample size calculation were previously published (reference no. 15, doi https://doi.org/10.3390/ijerph20032003 ). However, based on this suggestion, a brief description has been added to the method section of the reviewed manuscript.

  1. The authors need to present more robustly on the data collection procedure. Please clarify who was assigned to survey the study participants. Were they trained personnel or people with no medical background?

We thank the Reviewer for having stressed this point that gives us the opportunity to make some methodological clarification. The survey was an on-line, self-administered questionnaire; and the link redirecting to the questionnaire was shared by email. None of the research team was in charge of administering the survey, nor of collecting or verifying data. Actually, all the data were self-reported. The manuscript has been changed in order to improve clarity.

  1. The conclusion lacks the implications of the findings. Please make the necessary implications of your findings and discuss the implications of this study to reshape the residency training program.

We thank the Reviewer for having raised this point. Based on this comment, a brief paragraph has been added to the “Implications for Policy, Practice and Research” section of the current manuscript.

  1. There are numerous grammar and spelling errors throughout the manuscript that raise questions regarding the quality of the manuscript's preparation. Therefore, I request the authors screen for all grammatical and spelling errors and revise the manuscript text accordingly.

We thank the Reviewer for having highlighted these errors. The manuscript has been carefully proofread.

Reviewer 3 Report

Please explain the whether the use of a relatively small size of sample can generate convincing results.

Author Response

Response to Reviewer 3 Comments

  1. Please explain the whether the use of a relatively small size of sample can generate convincing results.

We thank the Reviewer for having raised this point. We performed a sample size calculation with the formula provided by Charan and Biswas. All data regarding sample size calculation were previously published (reference no. 15, doi https://doi.org/10.3390/ijerph20032003). However, based on this suggestion, a brief description has been added to the method section of the current manuscript. Considering the total number of the participants (379) and the calculated minimum sample size (315) our sample can be considered adequate.

Round 2

Reviewer 1 Report

Dear colleagues!

Thank you for your work and updates: now your research looks good and I’m satisfied in aswers.